# Carcass Traits of Growing Meat Goats Fed Different Levels of Hempseed Meal

**DOI:** 10.3390/ani12151986

**Published:** 2022-08-05

**Authors:** Reshma Gurung, Khim B. Ale, Frank W. Abrahamsen, Katie Moyer, Jason T. Sawyer, Nar K. Gurung

**Affiliations:** 1Department of Agricultural and Environmental Sciences, Tuskegee University, Tuskegee, AL 36088, USA; 2Kentucky for Hemp, Crofton, KY 42217, USA; 3Department of Animal Sciences, Auburn University, Auburn, AL 36849, USA

**Keywords:** carcass traits, goats, hempseed meal, marbling scores, ribeye area

## Abstract

**Simple Summary:**

Industrial hemp is currently being investigated as a potential new crop in the U.S. after the passage and approval of the 2014 and 2018 Farm Bills. Hemp plants grow efficiently, and its seeds are used in the production of hemp oil, leaving hempseed meal (HSM) as a byproduct, which is reported to be rich in crude protein (CP) around 30–38% on a dry matter basis and fiber, making it a possible feedstuff and a protein source for ruminants. However, limited work has been carried out to evaluate the effects of utilizing HSM as feedstuffs for goats on their carcass characteristics. This study aims to investigate the effects of feeding various levels of HSM on the carcass traits of the crossbred Boer goats. Results suggest that including up to 30% of HSM in the diet of growing meat goats has no adverse effects on their carcass traits and meat quality. These results might be encouraging for the hemp industry as HSM could potentially be marketed and used as an alternative protein source for livestock.

**Abstract:**

Hempseed meal (HSM) is the byproduct of hemp seeds and is rich in crude protein and fiber, making it an ideal candidate as a feedstuff for ruminants. The objective of the present study is to evaluate the effects of feeding different levels of HSM on the carcass traits of crossbred Boer goats. Forty castrated goat kids (approximately six months, 25.63 ± 0.33 kg) were assigned to one of four treatments (*n* = 10) in a completely randomized design. Goats were fed pelleted diets (50% forage and 50% concentrate) with additional supplementation of HSM: control with 0%, 10%, 20%, and 30% of the total diets. Goats were harvested and processed after a 60-day feeding trial. There were no significant differences (*p* > 0.05) in the mean values of dressing percentages, carcass weights, body wall thickness, and ribeye area among treatments. Marbling scores and percentages of moisture, fats, proteins, and collagen in the muscles showed no significant differences (*p* > 0.05) among the treatments. Results suggest that including up to 30% of HSM in the diet of growing meat goats does not affect their carcass traits.

## 1. Introduction

Industrial hemp (*Cannabis sativa* L.) is an annual herbaceous plant grown mainly for fibers, seeds, and various industrial products [1]. This variety of *Cannabis sativa* is legalized to cultivate as it produces 0.3% or less tetrahydrocannabinol (THC) [2]. However, other varieties of Cannabis, such as marijuana, contains 5–25% and shows more intoxicating and hallucinogenic properties [2]. Industrial hemp is currently being investigated as a potential new crop for livestock in the U.S. with the passage of the 2014 and 2018 Farm Bills in the USA [3,4] and legalized hemp production and cannabidiol (CBD) products derived from hemp [4]. Hemp plants thrive well, and its seeds are utilized in the production of hemp oil. The remaining byproduct is used in the production of hempseed meal (HSM), which is found to be rich in crude protein (CP), approximately 30 to 38% on a dry matter (DM) basis and fiber, making it an ideal candidate as a feedstuff for ruminant animals [1,5,6,7,8]. Hempseed oil is also abundant in essential fatty acids. It contains 50 to 70% linoleic acids and 15 to 25% alpha-linolenic acids [9]. It contains around 80% polyunsaturated fatty acids (PUFAs) and essential amino acids, especially arginine [5]. These polyunsaturated essential fatty acids, and easily digestible complete protein properties of hempseeds provide nutritional benefits to humans and animals [10]. It has an ideal omega 6 to omega 3 essential fatty acids ratio for optimal human health [5,11]. These polyunsaturated essential fatty acids can be utilized as another energy source for the animal while potentially improving the immune function [1]. On a DM basis, hempseed meal contains 32.08% CP, 50.79% neutral detergent fiber (NDF), 39.04% acid detergent fiber (ADF), 8.24% ash, and 5.24% ether extract [1]. Studies showed that HSM is an excellent source of rumen undegraded protein [1,5]. Gibb et al. [12] found no detrimental effect on the carcass traits of steers fed 14% full-fat hemp seed (HS). Hessle et al. [6] fed steers diets supplemented with cold-pressed hempseed cake and found no effect on the carcass traits compared to the soybean meal fed steers. There are few studies on the hempseed and its byproduct as a potential protein source for ruminants, especially goats [1,7,12]. 

Small ruminants such as goats can be vital in production arenas where nutritional resources are limited. Goats (*Capra hircus*) are small ruminants domesticated for meat, milk, fiber, and skins [13]. The meat goat industry is rapidly growing, and its demand and popularity is increasing in the US with the increase in the ethnic populations and immigrants from Asia, Africa, Latin America, and the Middle East [13]. Goat meat is considered healthier than other red meats [14] because it has leaner protein, less fat, and is high in iron and vitamin B12. It also has balanced amino acids, saturated/unsaturated fatty acids, low n6:n3 ratio, and high conjugated linoleic acid [15]. Carcass characteristics are important aspects of meat quality that help to determine the marketing value of the meats and live animals. So, it is highly significant to the producer and consumer. Meat goat carcass characteristics can be influenced by breed, age, sex, diet, and environment [16]. Numerous studies have been conducted to evaluate the effect of various agricultural byproducts on the different carcass characteristics of small ruminants [17,18,19]. Previous research evaluating the effect of HSM on fresh and cooked characteristics of meat are extremely limited. However, Smith et al. [20] reported no significant difference (*p* > 0.05) on the fresh and cooked characteristics in the goat meat fed with varying levels of HSM. Limited research has been undertaken on the carcass characteristics of goat meat [21]. So, the objective of the current study was to investigate the effects of feeding various levels of HSM on the carcass traits and the meat quality of crossbred Boer goats during the 60 days trial period.

## 2. Materials and Methods

### 2.1. Experimental Animals and Diets

The study was conducted at the Caprine Research and Education Unit of George Washington Carver Agricultural Experiment Station of Tuskegee University, Tuskegee, AL, USA. All animal handling, care, and sample collection procedures were conducted and approved by Tuskegee University Animal Care and Use Committee protocol number R07-2019-5. Goats were brought from Texas and quarantined for 14 days during which they were provided control complete total mixed ration diet. The goats were individually housed in 1.1 × 1.2 m pens with plastic-coated expanded metal floors. The goats were vaccinated with *Clostridium perfringens* type C and D-Tetani Bacterin-Toxoid (Bayer Corp., Shawnee Mission, KS, USA) and dewormed with Cydectin (moxidectin; Fort Dodge Animal Health, Fort Dodge, IA, USA) before arrival. Forty castrated Boer cross goats *(Capra aegagrus hircus)* with approximately six months of age and an initial average weight of 25.63 ± 0.33 kg were randomly allocated to one of four treatments (*n* = 10) in a completely randomized design for 60 days. Treatments consisted of different levels of HSM: control with 0%, 10%, 20%, and 30% HSM supplementation of the total diets. 

Hempseed meal was obtained from Kentucky Hemp Works, Crofton, KY, USA. Pelleted diets were prepared at Auburn University Poultry Feed Mill to ensure the goats consume as much HSM as possible by reducing the chance of selective feeding by goats. The complete diet consisted of bermudagrass (*Cynodon dactylon*) hay, soybean meal, meat maker 16:8 (goat premix), cracked corn, molasses, and HSM at varying rates. Goat diets consist of 50% concentrate mix (as-fed basis) and 50% mixed hay which were offered separately. The goats were fed pelleted diets formulated to meet their nutritional needs. Animals were fed twice a day with ad libitum access to water throughout the experiment. Additionally, each animal was provided with 0.23 kg of mixed hay to aid in optimal rumen function. Feed remnants were weighed twice daily and replaced to ensure animals had constant access to fresh feed.

### 2.2. Chemical Analysis 

The samples of hay and concentrate mixes were separated and collected before the experiment and dried for 48 hours at 55 °C in a convection oven (model 420, NAPCO, Pittsburgh, PA, USA). The samples were ground using a grinding machine (Hammer Mill Model 1250; Lorenz MFG Co., Benson, MN, USA) and sent to the Holmes laboratories, Millersburg, Ohio for analysis. Dry matter, crude protein, acid detergent fiber, crude fat, ash, calcium, and phosphorus were completed according to the methods described by the American Organization of Analytical Chemists [22]. The D.M. concentration of hay was measured at 60 °C for 24 hours, and ash was determined at a temperature of 550 °C for 5 hours. Nitrogen concentration (N) of the diet samples was determined using Kjeldahl N method, and CP was measured by: Crude Protein (CP)=Nitrogen (N)×6.25

Neutral detergent fiber (NDF) and acid detergent fiber (ADF) were calculated using the method of van Soest et al. [23] as utilized by the Ankom Fiber Analyzer (Ankom Technology Corp., Macedonia, NY, USA). Lignin concentration was determined according to methods described by the United States Department of Agriculture [24]. 

### 2.3. Slaughter and Carcass Evaluation 

On day 60 of the study, goats were weighed (final weight) utilizing a goat and sheep scale manufactured by Lakeland Farm and Ranch (Clawson, MI, USA) in 0.1 kg increments. Feed and water were withheld overnight before slaughter. The goats were transported and harvested according to the USDA standards at the Lambert-Powell Meats Laboratory, Auburn University, AL, USA. Goats were slaughtered using approved methods of US Department of Agriculture-Food Safety and Inspection Service (USDA FSIS) according to the Humane Slaughter Act [25]. After the slaughter process, the carcass was rinsed with hot water and a 2% solution of lactic acid and weighed using a Static Monorail Scale (Vandenberg Scales, Sioux Center, IA, USA) to determine the hot carcass weight (HCW). Kidney, pelvic, and heart (KPH) were collected to determine KPH fat percentage of the carcass weight using an analytical balance (PB3002-S, Met- tler Toledo, Columbus, OH, USA). Carcasses were then chilled at 4 °C for 24 hours, after chilling, carcasses were re-weighed using the Static Monorail Scale to determine the cold carcass weight (CCW). Dressing percentage (DP) was calculated by the following equation:$$Dressing\;Percentage\;\left(DP\right)=\left(\frac{Hot\;Carcass\;Weight}{Live\;Weight}\right)\times 100$$

Then, Ribeye area (REA) was determined by measuring the surface area of the longissimus dorsi muscle between the 12th and 13th ribs of the goat carcass using a grid [26]. Marbling scores were measured according to the USDA [27] utilizing beef marbling scores as reference, between the 12th and 13th rib interface. Fat thickness opposite to the loin eye was determined by measuring the subcutaneous fat over the ribeye area (REA) utilizing a caliper after measuring the REA. In addition, body wall thickness was measured approximately 4.5 inches from the midline of the ribbed goat carcass utilizing a stainless-steel ruler. Fresh meat samples were minced twice through a meat-grinding machine; minced meat samples were vacuum packaged and frozen for chemical analysis. The chemical composition of meat samples was determined by utilizing the proximate analysis to determine the moisture, fat, protein, and collagen content according to the Association of Official Analytical Chemists [28]. Samples for chemical analysis (protein, moisture, fat, and collagen) were conducted using a near-infrared (NIR) approved spectrophotometer (Food Scan™, FOSS Analytical A/S, Hilleroed, Denmark).

### 2.4. Statistical Analysis

Data were analyzed using the GLIMMIX procedures of SAS 9.4 (SAS Inst. Inc., Cary, NC, USA). Treatment served as the lone fixed effect for carcass measurements and physiochemical components. Least squares means were generated and statistical significance (*p <* 0.05), F-values were observed, and least squares means were separated using pair-wise t-tests (PDIFF option). 

## 3. Results and Discussion

### 3.1. Diet Composition 

The chemical composition of Bermuda grass hay (BGH) and hempseed meal (HSM) are shown in Table 1, while the ingredients and chemical composition of concentrate mixes are shown in Table 2. HSM is relatively high in CP concentration in the present study, and other studies reported similar values [1,6,7,8,29]. In this study, diets were balanced to be iso-nitrogenous replicating conditions that might be experienced in a production scenario. As the rate of HSM supplementation increased, there was a decrease in total digestible nutrients (TDN) and non-fiber carbohydrate (NFC), while there was an increase in crude fat. Fiber content, both ADF and NDF increased as the rate of supplementation increased. Lignin concentration, the indigestible portion of plant material, also increased with the increasing level of HSM. Phosphorus concentration also increased with an increasing level of supplementation while calcium decreased. As the level of HSM increased, net energy for gain (NEg) and TDN decreased, which could potentially induce a difference in growth and carcass characteristics; however, no significant differences were observed among treatments. 

### 3.2. Carcass Characteristics

The effect of different levels of HSM supplementation on carcass traits of goats are presented in Table 3 and Table 4. HSM supplementation showed no significant effect on the carcass weights (*p* > 0.05) evaluated in the present study (Table 3). There were no significant differences (*p* > 0.05) in the mean values of dressing percentages (DP) among treatments (Table 3). The values were 46.59, 45.42, 45.77, and 46.16% for diets containing 0, 10, 20, and 30% HSM, respectively, which were higher than that reported by Gurung et al. [18] when Dried Distillers Grain was used with soluble (DDGS) for goats. However, the DP was lower than that obtained when goats were supplemented with peanut skins [30], and grain diets and pasture [31]. The values of hot carcass weight (HCW) were 16.55, 16.37, 15.69, and 15.33 kg for diets containing 0, 10, 20, and 30% HSM, respectively, which are in decreasing order with increase of HSM supplementation but not significantly different (*p* > 0.05) among the treatments (Table 3). Similarly, the values of cold carcass weight (CCW) of the animals were decreasing with the increase in the levels of HSM supplementation but were not significantly different (*p* > 0.05) among the treatments (Table 3). The values of HCW and CCW of meat goats were higher than that reported by Ebrahimi et al. [32], whereas they were lower than that reported by Min et al. [30].

The marbling scores (376, 399, 355, 364, respectively) were also not significantly different (*p* > 0.05) among the treatments and fall on the marbling degree of “Slight (S.L.)”. Flank color and marbling scores were all within the normal range for growing meat goats, as these values are consistent with other values reported by other researchers [17,18,19,30,32]. Ribeye area (REA) was not significantly different (*p* > 0.05) among treatments with values of 3.68, 3.4, 3.47, 3.39 cm^2^ for 0, 10, 20, and 30% HSM, respectively. There were no differences in the body wall (BW) thickness (*p* > 0.05) among treatments (Table 3), with 0.31, 0.30, 0.29, and 0.29 cm for 0, 10, 20, and 30 % HSM, respectively. The results were lower than those of Gurung et al. 2009 [18], reported the BW thickness of 1.09 cm for goats fed with 10.3 % DDGS and 3.9 cm BWF thickness reported by 50 % peanut skins (PS) supplementation on the goats [30]. Similarly, back fat thickness measured opposite loin eye was not significantly different (*p* > 0.05) among treatments. The values were 0.033, 0.027, 0.032, and 0.025 cm for diets containing 0, 10, 20, and 30% HSM, respectively. The results were lower than the result of 0.8 cm of fat depth for 30% PS supplementation presented by Min et al. [30]. The KPH fat percentage was not affected (*p* > 0.05) by varying levels of HSM supplementation in the meat goats. These results suggest that varying levels of HSM supplementation have no negative impact on the carcass characteristics of goat meat.

The results reported by Gurung et al. [18] also agreed with the results of the present study that varying levels of agricultural protein by products do not significantly impact carcass characteristics. Furthermore, Abdelrahim et al. [17] fed varying levels of DDGS (12.7 and 25.4% of diet) to growing lambs and determined that the different levels of DDGS had no negative impact on carcass characteristics evaluated in that study. There was no significant difference (*p* > 0.05) in the moisture, fat, protein, and collagen content of the goat meats among treatments, as shown in Table 4. The percentage of moisture, fat, protein, and collagen were all within the normal range for growing meat goats, consistent with the values reported by other studies [17,18,19,30,32]. Similarly, moisture and protein contents were consistent with the values reported by Sen et al. and Shija et al. [31,33]. However, the fat percentage of goat meat was higher than that reported by previous researchers [31,33]. These results showed that including up to 30% HSM supplementation has no detrimental effects on the physio-chemical attributes of goat meat.

## 4. Conclusions

Limited studies have been implemented regarding the effect of supplementing HSM on the carcass traits and meat quality of growing meat goats. So, these results suggest that up to 30% HSM can be fed to growing meat goats without affecting the carcass traits. HSM could potentially be an effective protein source for growing meat goats since even the 30% level has no adverse effects. These results might be promising for the future of the industrial hemp industry as HSM could potentially be marketed as an alternative protein source for livestock. Further works are needed to evaluate the effects of HSM inclusion rates on the carcass traits and meat quality of goats. 

## Figures and Tables

**Table 1 animals-12-01986-t001:** Chemical composition of Bermuda grass hay (BGH) and hempseed meal (HSM) used in the experiment.

^a^ Items	Diet	
	BGH	HSM
Dry Matter (%)	95.85	89.61
Crude Protein (%)	9.58	36.42
Acid detergent fiber (%)	42.82	36.47
Neutral detergent fiber (%)	70.99	49.47
Lignin (%)	6.68	12.76
Crude Fat (%)	1.23	11.53
Total digestible nutrients (%)	55.13	63.22
Net energy for gain (Mcal/lb)	0.266	0.377
Ash (%)	4.79	5.82
Calcium (%)	0.26	0.23
Phosphorus (%)	0.17	1.03

^a^ Values are presented on a dry matter basis, except dry matter. Source: Holmes Laboratory Inc., Millersburg, OH 44654, USA.

**Table 2 animals-12-01986-t002:** Ingredients and nutrient composition of the experimental diets fed to Boer crossbred meat goats.

Items ^a^	Diets			
	0% HSM	10% HSM	20% HSM	30% HSM
Ingredients, as-fed basis				
Bermuda Grass Hay (%)	50	50	50	50
Cracked Corn (%)	28	24	18.5	14
Soybean Meal (%)	18.5	13	8	2.5
Molasses (%)	2.5	2.5	2.5	2.5
^b^ Goat Premix (%)	1	1	1	1
Nutrient composition, DM basis				
Dry Matter (%)	89.06	88.66	89.1	89.86
Crude Protein (%)	19.18	19.91	19.25	20.39
Lignin (%)	3.34	4.77	6.21	7.02
Acid Detergent Fiber (%)	21.06	24.67	28.96	30.97
Neutral Detergent Fiber (%)	33.29	35.18	39.63	42.8
Non-Fiber Carbohydrate (%)	40.64	37.8	35.38	31.9
Acid Hydrolysis Fat (%)	3.19	3.32	4.22	4.5
Total Digestible Nutrients (%)	71.2	69.2	64.7	62.79
Net Energy for gain (Mcal/lb)	0.481	0.456	0.397	0.372
Ash (%)	7.02	7.01	7.09	6.79
Calcium (%)	0.95	0.92	0.88	0.82
Phosphorus (%)	0.39	0.41	0.48	0.52

^a^ Values are presented on a dry matter basis, except DM. ^b^ Goat premix (%): Ca 9.0, P 8.0, salt 41.0, K 0.1, Mg 1.0; (ppm) Cu 1750, Se 25.0, Zn 7500 and (IU kg^−1^) Vitamin A 3,08,644, Vitamin D 24,251, and Vitamin E 1653.

**Table 3 animals-12-01986-t003:** Effects of varying levels of HSM supplementation on the carcass yields and quality measurements of meat goats.

Items	Treatment, %				SEM	*p*-Value
	0	10	20	30		
Dressing Percent, %	46.59	45.43	45.77	46.17	0.510	0.107
Hot carcass weight, kg	16.55	16.37	15.69	15.33	0.42	0.153
Cold carcass weight, kg	16.42	16.09	15.51	15.15	0.418	0.152
Ribeye Area, cm^2^	3.68	3.40	3.47	3.39	0.119	0.297
Fat Thickness Opposite Loin Eye, cm	0.033	0.027	0.032	0.025	0.004	0.556
Body Wall Thickness, cm	0.31	0.30	0.29	0.29	0.036	0.975
Kidney, Pelvic, Heart Fat, %	2.20	2.70	2.60	2.45	0.295	0.656
Marbling score *	376	399	355	364	16.5	0.278

* Means based on USDA (2001) marbling scores where 100 = practically devoid (Standard^−^), 200 = traces (Standard^+/0^), 300 = slight (Select^+/−^), 400 = small (Choice^−^), 500 = modest (Choice^0^), 600 = moderate (Choice^+^), 700 = slightly abundant (Prime^−^) and 800 = moderately abundant (Prime^0^). SEM = standard error of mean.

**Table 4 animals-12-01986-t004:** Physio-chemical attributes of goat meats supplemented with varying levels of HSM.

Meat Quality Traits	Treatment, %				SEM	*p*-Value
	0	10	20	30		
Moisture, %	71.28	72.82	71.24	73.58	0.757	0.087
Fat, %	13.69	11.49	13.68	10.18	1.069	0.065
Protein, %	22.95	22.96	23.09	23.72	0.383	0.446
Collagen, %	2.53	2.31	2.52	2.29	0.075	0.056

SEM = standard error of mean.

## Data Availability

Data available upon reasonable request from the corresponding author.

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
