# Peer review of "Carcass Traits of Growing Meat Goats Fed Different Levels of Hempseed Meal"

_animals, 2022, doi:10.3390/ani12151986_

Round 1
Reviewer 1 Report
Dear gentlemen;
The work deserves credit and we think it shiuld be published with some little changes, as
follows:
Calcium is mentioned in the text, in table 1 and 2 as quantified, but in item 2.2 chemical
analysis, it is not described how it was done.
In table two, lignine quantity, review this mentioned quantity, in the 30% HMS diet.
Expand comments:
1 What is the main question addressed by the research?
The research aims to investigate the effects of feeding various levels of Hempseed Meal on the carcass traits and the meat quality of cross-bred Boer goats during the 60 days trial period.
2 Do you consider the topic original or relevant in the field, and if so, why?
The coproduct of hempseed free of tetrahydrocannabinol (Cannabis sativa) used with goats is not unique, but the approach regarding quantities seems to be another complementary work to these studies. After the oil extraction, the hempseed wheat contains proteins, high levels of soluble and insoluble dietary fibers and essential amino acids. We think the effect of different levels of HMs relevant to carcass trait and meat.
3 What does it add to the subject area compared with other published
material?
There are several studies on the use of HSM in animal nutrition, as this is a great source of fiber, vegetable protein, omega-3 and omega-6 fatty acids, in addition to being an important source of minerals. As far we know, there are papers on use of HSM in cultures of crossbred Boer goats, but we believe that further studies are necessary to determine the safety and also the optimal levels of supplementation, and due to its high nutritional value, it is worth researching its use as a food supplement. A work worth mentioning is that of Abrahamsen et all (2021) (https://doi.org/10.15232/aas.2021-02153) which uses hempseed and analyzes ruminal fermentation among several parameters. In the present study, carcass and meat quality was evaluated for 60 days fed with Hempseed meal
4 What specific improvements could the authors consider regarding the
methodology
Rumen evaluation and determination of other metabolic parameters such as rumen volatile fatty acid profile and energy parameters, glucose, blood NEFA to see the influence of HSM inclusion on metabolism, carcass quality and meat tenderness.
5 - Are the conclusions consistent with the evidence and arguments
presented and do they address the main question posed
Yes, the main question is addressed, which is the quality of meat and carcass traits of goats feed with different quantities of HSM.
6 Please include any additional comments on the tables and figures
Calcium is mentioned in the text, in table 1 and 2 as quantified, but in item 2.2 chemical
analysis, it is not described how it was done.
In table two, lignin quantity, review this mentioned quantity, in the 30% HMS diet.
Author Response
Response to Reviewer-1 Comments
Thank you for editing and for the questions of clarity regarding the submitted manuscript. We have completed and compiled tremendous changes wholistically to the entire submission based on your comments. We are grateful for these improvements to the presentation of our work and believe that these changes were necessary. We hope that the clarification below and within the text of the submission have completed all the necessary steps to achieve acceptance for publication.
Point 1: Calcium is mentioned in the text, in table 1 and 2 as quantified, but in item 2.2 chemical analysis, it is not described how it was done.
Response 1: Calcium was also determined according to the methods described by the American Organization of Analytical Chemists. Calcium has been added to the body of the text per the reviewer comment.
Point 2: In table two, lignin quantity, review this mentioned quantity, in the 30% HMS diet.
Response 2: Yes, the lignin quantity was a mistake, and it is corrected per the reviewer comment.

Reviewer 2 Report
- In the introduction, mention the possible hallucinogenic properties of cannabis;
- There was no transition period (7-14 days) in goat fattening;
- The method for measuring REA was not given;
- The header of table 1 says "Concentrate" and it should be HSM.
-Can Hempseed Meal (HSM) be used in fattening goats without adversely affecting the fattening results?
-The topic is important because the use of waste feed improves profitability.
-It has not always been confirmed that HSM can be used in animal nutrition. The results obtained so far concern mainly cattle.
-The conclusions correspond to the set goal.
-The appropriate statistical test was applied. The tables are easy to read.
Author Response
Response to Reviewer-2 Comments
Thank you for editing and for the questions of clarity regarding the submitted manuscript. We have completed and compiled tremendous changes wholistically to the entire submission based on your comments. We are grateful for these improvements to the presentation of our work and believe that these changes were necessary. We hope that the clarification below and within the text of the submission have completed all the necessary steps to achieve acceptance for publication.
Point 1: In the introduction, mention the possible hallucinogenic properties of cannabis.
Response 1: Information regarding possible hallucinogenic properties of cannabis was added to the body of the text per the reviewer comment.
Point 2: - There was no transition period (7-14 days) in goat fattening.
Response 2: Goats were acclimated to the research site during the quarantine period. During the acclimation period, goats were provided complete total mixed ration diet consisting of timothy hay, soybean meal, meat maker 16:8 (goat premix), corn and molasses without the inclusion of HSM.
Point 3: - The method for measuring REA was not given;
Response 3: It is corrected per the reviewer comment.
Nash, S. Using grids to determine carcass measurements. Available online: https://www.uidaho.edu/-/media/UIdaho-Responsive/Files/Extension/4-H/Animal-Science-Lesson-Plans/using-grids-to-determine-carcass-measurements.pdf?la=en&hash=11B92BB0D9851A7BD9AEE68517EA1D6AE51B424A (accessed on 26 July 2022).
Point 4: - The header of table 1 says "Concentrate" and it should be HSM.
Response 4: It is corrected per the reviewer comment.
Point 5: Can Hempseed Meal (HSM) be used in fattening goats without adversely affecting the fattening results?
Response 5: Literature supporting the inclusion of HSM in diets, specifically ruminants is limiting. The presentation of results within the current study are some of the first to be presented. The current results indicate that there are no deleterious impacts of including HSM within small ruminants specifically goats.

Reviewer 3 Report
Comments and Suggestions for Authors
I would like to thanks Authors for proposing a very interesting study. It is better for the authors to use more concise language. In the manuscript, some language editing is required to correct the grammar and syntax errors. Some expressions need to be more precise and concise to avoid confusing readers.
After reading the work, there remains a certain lack of information. How different levels of HSM affected growth performance, carcass characteristics, oxidative stability of meat, pH...This study could be more elaborated to give a more complete picture of the effect of HSM additive on goat carcass quality.
Simple Summary
Line 11-14: It is a very long sentence that confuses the reader. Please, divide and rewrite the sentence.
Abstract
Line 28-31: If there are no statistical differences (p > 0.05), do not provide individual data in Abstract.
Introduction
Why was goat selected as the model organism for this study? There is no information in the introduction about the goat, the quality and properties of the goat meat. Please complete the “Introduction" section with this information.
Line 39-40: “as it differs from marijuana, which produces 0.3 % or less tetrahydrocannabinol (THC) than that of marijuana” Please, rewrite. Use synonyms to make the sentence readable.
Line 43-47: It is a very long sentence that confuses the reader. Please, divide and rewrite the sentence.
Materials and Methods
Line 126-127: Were the analyzes performed on frozen material? How are the meat samples prepared for analysis? Was it dried or freeze-dried?
Line 134-140: Marbling scores can be presented in the table, the text will be easier to read.
Line 144-145: How kidney, pelvic, and heart fat were collected to avoid measurement error? Please, add methodology.
Line 150: What statistical test was used to analyze the data?
Results and Discussion
The results have been presented correctly. However, the discussion is very modest and needs to be expanded.
Author Response
Response to Reviewer – 3 Comments
Thank you for editing and for the questions of clarity regarding the submitted manuscript. We have completed and compiled tremendous changes wholistically to the entire submission based on your comments. We are grateful for these improvements to the presentation of our work and believe that these changes were necessary. We hope that the clarification below and within the text of the submission have completed all the necessary steps to achieve acceptance for publication.
Point 1:After reading the work, there remains a certain lack of information. How different levels of HSM affected growth performance, carcass characteristics, oxidative stability of meat, pH...This study could be more elaborated to give a more complete picture of the effect of HSM additive on goat carcass quality.
Response 1: The authors concur with reviewer comments that the information was limited. Results presented provide some of the first data to support use of HSM in small ruminants, specifically goat diets as a feed by-product. Current studies investigating HSM in cattle does not aid in the presentation of the results of the current study. Additional research is needed to provide greater clarity on the inclusion of the HSM in ruminant diets both small and large.
Point 2: Line 11-14: It is a very long sentence that confuses the reader. Please, divide and rewrite the sentence.
Response 2: It is corrected per the reviewer's comment.
Point 3: Line 28-31: If there are no statistical differences (p > 0.05), do not provide individual data in Abstract.
Response 3: It is corrected per the reviewer's comment.
Point 4:Why was goat selected as the model organism for this study? There is no information in the introduction about the goat, the quality and properties of the goat meat. Please complete the “Introduction" section with this information.
Response 4: Additional purpose for use of goats has been included into the body of text per the reviewer's comment.
Point 5: Line 39-40: “as it differs from marijuana, which produces 0.3 % or less tetrahydrocannabinol (THC) than that of marijuana” Please, rewrite. Use synonyms to make the sentence readable.
Response 5: It is corrected per the reviewer's comment.
Point 6: Line 43-47: It is a very long sentence that confuses the reader. Please, divide and rewrite the sentence.
Response 6: It is corrected per the reviewer's comment.
Point 7: Line 126-127: Were the analyzes performed on frozen material? How are the meat samples prepared for analysis? Was it dried or freeze-dried?
Response 7: Information regarding sample analysis was added to the body of the text per the reviewer's comment.
Point 8: Line 134-140: Marbling scores can be presented in the table, the text will be easier to read.
Response 8: The authors concur with reviewer comments. It is corrected per the reviewer's comment.
Point 9: Line 144-145: How kidney, pelvic, and heart fat were collected to avoid measurement error? Please, add methodology.
Response 9: Information regarding the measurement of KPH was added to the body of the text per the reviewer's comment.
Point 10: Line 150: What statistical test was used to analyze the data?
Response 10: Statistical analysis has been revised for clarity per the reviewer's comment.
Point 11: The results have been presented correctly. However, the discussion is very modest and needs to be expanded
Response 11: Some points on the discussions are added to the body of the text per the reviewer's comment.
